# Navigating Evolving Challenges in Blood Safety

**DOI:** 10.3390/v16010123

**Published:** 2024-01-15

**Authors:** Mahmoud Reza Pourkarim

**Affiliations:** 1Laboratory for Clinical and Epidemiological Virology, Rega Institute for Medical Research, Department of Microbiology, Immunology and Transplantation, KU Leuven, Herestraat 49, 3000 Leuven, Belgium; mahmoudreza.pourkarim@kuleuven.be; 2Health Policy Research Centre, Institute of Health, Shiraz University of Medical Sciences, Shiraz 71348-14336, Iran; 3Blood Transfusion Research Centre, High Institute for Research and Education in Transfusion, Tehran 14665-1157, Iran

Blood safety remains a paramount public health concern, and health authorities maintain a high level of vigilance to prevent transfusion-transmitted infections (TTIs) [1]. Rigorous policies, including healthy donor selection, pathogen detection, and pathogen reduction in donated blood, are crucial post-donation safety measures [2]. Historically, the universal implementation of enzyme immunoassays marked the first generation of blood screening tests in modern blood banks, significantly reducing the transmission of blood pathogens. This approach played a crucial role in revealing the prevalence of bloodborne pathogens in both general populations and high-risk groups [3]. However, the adoption of nucleic acid tests (NATs) in blood screening units has been transformative, markedly enhancing the specificity and sensitivity of pathogen detection tests [4]. Consequently, the identification of infected donors was dramatically elevated, and the risk of transfusion-transmitted infections (TTIs) significantly decreased [5]. Moreover, molecular techniques have aided in categorizing viral strains into distinct groups and subgroups, providing precise insights into the geographical distribution of circulating bloodborne pathogens [6,7,8,9]. Additionally, the application of these molecular techniques has facilitated the development and production of vaccines and antiviral medications, resulting in a significant decrease in the carrier population of bloodborne pathogens [10]. These improvements have also assisted health authorities in implementing tailored elimination programs for some of these pathogens, customized for each country [11,12,13,14].

In addition to implementing applied strategies for blood safety, it is crucial to have a comprehensive understanding of the epidemiology of bloodborne pathogens and the biotic and abiotic factors that influence it [15,16]. This understanding is essential for delineating the panel of blood screening tests in local blood banks based on the prevalence of bloodborne pathogens in a specific geographical region. It appears that recent global changes are likely to drive a series of challenges and disrupt the longstanding endemic boundaries [17].

Climate change has led to alterations in both temperature and precipitation averages at regional and global levels over the course of several years [18]. As a result of ambient events, such as droughts, flooding, and forest fires, ecosystem conditions and, consequently, their inhabitants have undergone modifications. Human and animal populations are relocating from regions affected by natural disasters and settling in industrialized or often newly deforested locations [19,20,21]. Following this, in the newly opened-up ecosystem, pathogens are adapting to new vectors and intermediate hosts, leading to the emergence of new outbreaks [17,22]. Additionally, travel to or from this newly emerged ecosystem may establish new epidemiological corridors and enhance the dispersal of bloodborne infections [23,24].

Indeed, higher temperatures and increased humidity frequently promote the abundance of vectors [25,26]. Furthermore, the intricate relationship between humidity and temperature, encompassing variations in their levels and durations (increase, decrease, prolongation, and shortening), significantly impacts the winter survival of arthropods [27], which directly speed up the vector–host interaction, the host biting rate, the frequency of pathogen proliferation, and, finally, the life cycle complementation of vector-borne blood pathogens [28,29].

A recent study examining the impact of climate change forecasts on West Nile Virus (WNV) expansion reveals a concerning fivefold increase in the risk of WNV infection in Europe over the coming decade. Remarkably, Western Europe could potentially face significant outbreaks, irrespective of the extent of climate change [30]. Additionally, the same modeling predicts more frequent outbreaks of the dengue virus in the future in Pakistan [31].

Public health interventions traditionally focus on vector control. However, natural disasters significantly impact these measures, exposing humans to vectors. To address this, it is crucial for blood bank authorities to establish an intelligent surveillance system that includes a risk assessment of bloodborne pathogens, accounting for local climate variables. This system, relying on environmental parameters, can assess the risk of bloodborne infections in specific geographical areas and accordingly update the blood screening assay panel. Drawing from lessons learned in recent pandemics, a preparedness program can assist in preventing or responding to pathogen outbreaks caused by climate change. An investigation of pathogen traces in sewage, a valuable lesson from the COVID-19 pandemic, has led to the establishment of a surveillance system for various threatening pathogens [32].

Human mobility, animal trades, and migrations fall under the second category of challenges in the field of transfusion medicine. These activities have expedited the relocation and dissemination of pathogens and their associated vectors more rapidly and efficiently than in the past [33]. Historically, political or economic instabilities and, more recently, domestic conflicts, regional violence, and persecution, plus natural disasters, have served as “push factors”, compelling populations to immigrate and seek new opportunities in more developed countries. In many cases, immigrants originate from geographical areas that are endemic for certain communicable diseases or where the prevalence of infectious diseases is significantly higher than in the destination countries [34,35,36,37]. It is worth noting that, for various reasons, individuals in these populations have limited access to health services, including screening, vaccination, and appropriate treatment. Moreover, the prolonged process of immigration can heighten the risk of infectious disease outbreaks among these marginalized populations, potentially impacting the prevalence of specific infectious diseases in the destination areas [38,39,40,41]. Several studies have demonstrated a notable prevalence of bloodborne diseases, including neglected tropical diseases, within these communities. It is imperative to establish a healthcare system to provide essential healthcare amenities for these populations [42,43]. Establishing a protective system at immigrant reception camps, involving the identification of infected individuals, treatment, screenings, and subsequent vaccinations, could contribute significantly to mitigating potential threats to blood safety [44].

The third challenge to blood safety stands distinct from the two aforementioned challenges. Based on a provided list of bloodborne pathogens by the American Association of Blood Banks (AABB), a total of 68 microorganisms have been identified that can be transmitted through blood and blood products [45]. However, unexpected bloodborne pathogens and emerging infectious disease (EID) agents remain significant concerns for blood safety. Dengue virus, Chikungunya virus, and, most recently, Zika virus are examples of recently emerged infectious diseases [46]. This highlights an urgent need for innovative strategies or techniques in the surveillance and discovery of previously unforeseen and potentially harmful pathogens that could jeopardize blood transfusion safety. This urgency has been effectively addressed by the high-throughput sequencing technique (HTS).

The capacity of HTS, particularly next-generation sequencing (NGS), to sequence the complete metagenome of biological samples has revolutionized the diagnostic field. Furthermore, this technique, coupled with other advanced disciplines, such as proteomics, transcriptomics, and metabolomics, and integrated with computational methodologies, is applied to detect and investigate all microorganisms at their community levels [47,48]. This technique has been extensively employed to spotlight the microbiome, particularly the bacteriome, in various ecological niches of the human body, with a particular emphasis on the human gut. The discovery of numerous associations between gastrointestinal, skin, or vaginal bacteria (microbiome) and a variety of illnesses is a direct outcome of the deployment of this technique in clinical investigations [49]. Viral communities, referred to as the virome in these body compartments, exert an indirect influence on these associations by impacting bacterial populations, either increasing or decreasing them [50]. However, armed with NGS, the human microbiome project (HMP) has primarily focused on the bacteriome.

NGS has added significant value to diagnostic virology by identifying minor or major viral populations in various ecological niches of animals, insects, or human bodies. This has addressed the hidden threat of unanticipated viruses in different human illnesses. The introduction of astrovirus MLB2 as the cause of febrile disease and meningitis [51,52], the identification of a new Bunyavirales virus as the cause of thrombocytopenia and leukopenia illness in China [53], the association of novel rhabdoviruses with acute hemorrhagic fever [54], and the identification of polyomavirus as the etiologic agent for human Merkel cell carcinoma [55] are a few instances showcasing the remarkable impact of this technology in clinical virology. In light of these accomplishments, the identification and characterization of viral swarms present in the blood (blood virome) represent another intriguing facet of the application of NGS.

NGS allows for a comprehensive exploration of the entire viral landscape in blood. While we previously believed blood to be sterile, recent studies have estimated the presence of approximately 10^5^ viral particles per milliliter [56]. By using NGS, Giant Blood Marseillevirus, human pegivirus 1 (HpgV-1), papillomavirus, and gemycircularvirus, as well as members of Picornaviridae, Circovirdae, and Astroviridae, could be detected in the blood stream [57,58,59,60,61]. Furthermore, through the application of this method, endogenous retroviruses, which are linked to several neurological, inflammatory, and infectious diseases, become detectable in the bloodstream [62,63,64,65]. Additionally, by relying on NGS, the translocation of prokaryotic viruses from the gut to the blood in certain pathological disorders has been traced [66]. The significance of the blood virome is underscored by the fact that, unlike other body compartments, blood is a systematically closed tissue. It lacks connections to the external environment and does not harbor commensal bacteria. Therefore, commensal viruses in the blood, such as HpgV and Anelloviruses, can directly interact with host cells. These viruses may influence the clinical outcomes of illnesses, and the clinical treatment of patients can, in turn, impact their population or evolution.

However, similar to the virome composition of other body compartments, the blood virome is susceptible to interventions. It has been demonstrated that the virome plays a pivotal role in human health, and disturbing this community exerts a detrimental impact. Factors such as vaccination, infection, and the administration of therapeutic or immunomodulatory agents, as well as solid or hematopoietic transplantations, have the potential to disrupt the composition of the blood virome [66,67,68]. The modulations induced by these interventions could introduce permanent changes and might lead to various illnesses [69]. Our recent investigation on the most prevalent member of the blood virome, the Anellovirus family, shows that the administration of immunosuppressive agents to liver transplant recipients [70], antiviral therapy in HBV carriers [71], multiple blood transfusions in patients with blood disorder [72], and convalescent plasma therapy in COVID-19 patients [73] has an impact on the conformation of the blood virome. Notably, there is a trace of evidence indicating a potential detrimental effect of certain Anellovirus subgroups in specific conditions, both independently and in co-infection with certain pathogenic viruses [74,75,76]. Although we have not identified a driving role for Anelloviruses, they might not be merely considered bystanders.

To respond to the aforementioned three challenges, we need to establish a new paradigm in transfusion safety. These strategies should simultaneously tackle the challenges posed by global warming and human/animal immigration as potential sources of the emergence or re-emergence of bloodborne pathogens. An urgently needed component is a well-coordinated monitoring system of climate parameters linked to the favorability of pathogens and vectors. This real-time climate-based risk assessment can be complemented by deferring donations from individuals who have recently traveled to newly endemic regions. Additionally, to minimize the perturbations of personalized blood viromes, urging a more sparing use of prescriptions and blood products can be useful.

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
