# Peer review of "Navigating Evolving Challenges in Blood Safety"

_viruses, 2024, doi:10.3390/v16010123_

Round 1

Reviewer 1 Report

Comments and Suggestions for Authors

Accept in the current format.

Typically, in a standard editorial, an expert editor takes the initiative to highlight various challenges that researchers in the field can consider for their research articles or reviews. The author/editor of this manuscript is an expert and has shed light on three critical topics representing challenges in blood safety (the theme of your special issue).

The primary focus by the research is on the most significant challenges in blood safety. Absolutely, the author has addressed relevant questions and identified existing gaps in the field. This manuscript has extracted crucial points in blood safety, with some of them being uniquely emphasized in the literature.

Author Response

Dear Editor and Referees,

Thank you very much for considering my editorial (viruses-2805527), entitled “Navigating Evolving Challenges in Blood Safety” for publication in the Viruses. I greatly appreciate the comments given by the referees that helped us to improve the manuscript. In the revised version, I addressed issues raised by the reviewers. Accordingly, the title was changed to “Editorial”, and the references list was improved by updating some references of the text. Also couple of sentences plus related references were added to the introduction. These sentences have been highlighted.

These improvements have also assisted health authorities in implementing tailored elimination programs for some of these pathogens, customized for each country [11-14].

I hope that my manuscript aligns with the publication criteria of Viruses. You can find the revised version of the manuscript, complete with highlighted changes, in the system.

Yours sincerely,

Author: Mahmoud Reza Pourkarim.

Reviewer 2 Report

Comments and Suggestions for Authors

sent in an attached file

Author Response

Dear Editor and Referees,

Thank you very much for considering my editorial (viruses-2805527), entitled “Navigating Evolving Challenges in Blood Safety” for publication in the Viruses. I greatly appreciate the comments given by the referees that helped us to improve the manuscript. In the revised version, I addressed issues raised by the reviewers.

Accordingly, the title was changed to “Editorial”, and the references list was improved by updating some references of the text. Also couple of sentences plus related references were added to the introduction. These sentences have been highlighted.

"These improvements have also assisted health authorities in implementing tailored elimination programs for some of these pathogens, customized for each country [11-14]"

I hope that my manuscript aligns with the publication criteria of Viruses. You can find the revised version of the manuscript, complete with highlighted changes, in the system.

Yours sincerely,

Author: Mahmoud Reza Pourkarim.

Reviewer 3 Report

Comments and Suggestions for Authors

Thanks for this interesting Editorial section. It was a pleasure to review on this theme. I have only minor comments.

Author Response

Dear Editor and Referees,

Thank you very much for considering my editorial (viruses-2805527), entitled “Navigating Evolving Challenges in Blood Safety” for publication in the Viruses. I greatly appreciate the comments given by the referees that helped us to improve the manuscript. In the revised version, I addressed issues raised by the reviewers. Accordingly, the title was changed to “Editorial”, and the references list was improved by updating some references of the text. Also couple of sentences plus related references were added to the introduction. These sentences have been highlighted.

These improvements have also assisted health authorities in implementing tailored elimination programs for some of these pathogens, customized for each country [11-14].

We hope that my manuscript aligns with the publication criteria of Viruses. You can find the revised version of the manuscript, complete with highlighted changes, in the system.

Yours sincerely,

Author: Mahmoud Reza Pourkarim